# Simulation and Analysis of Imaging Process of Phosphor Screens for X-Ray Imaging of Streak Tube Using Geant4-Based Monte Carlo Method

**DOI:** 10.3390/s25030881

**Published:** 2025-01-31

**Authors:** Zichen Wang, Riyi Lin, Yuxiang Liao, Lin Tang, Zhenhua Wu, Diwei Liu, Renbin Zhong, Kaichun Zhang

**Affiliations:** 1School of Electronic Science and Engineering, University of Electronic Science and Technology of China, Chengdu 611731, China; 202322021009@std.uestc.edu.cn (Z.W.); 202421020201@std.uestc.edu.cn (R.L.); 202222021409@std.uestc.edu.cn (Y.L.); 202122022110@std.uestc.edu.cn (L.T.); wuzhenhua@uestc.edu.cn (Z.W.); dwliu212220@163.com (D.L.); rbzhong@uestc.edu.cn (R.Z.); 2Terahertz Radiation and Application Key Laboratory of Sichuan Province, University of Electronic Science and Technology of China, Chengdu 611731, China

**Keywords:** CCD sensor, ultrafast diagnostic technology, Monte Carlo method, streak tube

## Abstract

Ultrafast diagnostic technology has caused breakthroughs in fields such as inertial confinement fusion, particle accelerator research, and laser-induced phenomena. As the most widely used tool for ultrafast diagnostic technology, investigating the characteristics of streak cameras in the imaging process and streak tubes’ complex physical processes is significant for its overall development. In this work, the imaging process of a streak camera is modeled and simulated using Geant4-based Monte Carlo simulations. Based on the selected phosphor screen P43 (Gd_2_O_2_S: Tb) and charged coupled device (CCD) sensor parameters, Monte Carlo simulation models of phosphor screens and CCD sensors (We refer to the sensor parameters of the US company onsemi’s KAF-50100 sensor, but some adjustments are made during the simulation), implemented with the toolkit Geant4, are used to study the electron beam to generate fluorescence on phosphor and photoelectrons on CCD sensors. The physical process of a high-energy electron beam hitting a phosphor screen and imaging on the CCD camera is studied. Meanwhile, merits such as the luminous efficiency of the selected phosphor, spatial resolution of the phosphor screen, and spatial resolution of the selected CCD sensor are analyzed. The simulation results show that the phosphor screen and CCD sensor simulation models can accurately simulate the selected components’ performance parameters with the imaging process’ simulation results precisely reflecting the distribution of output electrons in the streak image tube. References for simulation and device selection in the subsequent research on streak cameras can be provided.

## 1. Introduction

Numerous-in-depth studies on matter’s basic structure and principles, such as the formation of chemical bonds, quantum tunneling, photoionization phenomena, and laser-induced phenomena, belong to the category of ultrafast phenomena [1]. Ultrafast phenomena are mysteries as these are usually difficult to be observed by conventional technology. Therefore, ultrafast diagnostic technology, with an ultra-high temporal and spatial resolution to capture and image ultrafast processes, recording images that cannot be captured by the human eye, has emerged to detect and observe ultrafast phenomena and has made remarkable progress [2,3]. Recently, ultrafast diagnostic technology has become the most important technique for studying the dynamics of microscopic particles in matter, as it can clearly present high-resolution imaging, significantly expand ordinary understanding of the mechanisms of things, and reveal more previously obscured phenomena and processes. New scientific discoveries and fields can then be expected and created, bringing more innovation and progress in science and technology.

As a widely used ultrafast diagnostic tool, the time resolution of a streak camera [4,5,6,7,8,9,10,11], with its vital device streak tube, is in fact a type of vacuum device [12,13]. The streak camera can display the intensity information of the observed points at different times along the vertical direction to form a streak. The streaks of several observed points can be merged together to form a streak camera photo.

The phosphors, being the key material of the phosphor screen for streak tubes, monitors, image intensifier and others, can transform the electronic images into visible optical images. Performances of the phosphor screen greatly influence on the resolution, luminescence spectrum, modulation transfer function and others. Given the current development of phosphors, several performance studies [14,15,16,17,18,19] were carried out on phosphors of P20 ((Zn, Cd) S: Ag), P22 (ZnS: Cu, Al), P31 (ZnS: Cu), P43 (Gd_2_O_2_S: Tb) and P45 (Y_2_O_2_S: Tb) for the phosphor screen of imaging.P20 and P22 are phosphors produced by Nichia in Japan, and P31, P43 and P45 are phosphor produced by Phosphor-Technology in Britain.

The study on streak cameras’ imaging process is critical to the research of streak cameras. The process can be decomposed into three stages: the physical process of electrons impinging onto the phosphor screen and generating photons, the process of photons transporting in the optical fiber panel, and the process of photons illusion on the CCD sensor and to excite secondary electrons. By recording photoelectrons, spatial information of the target or detecting object can be traced and analyzed. In this study, the optical system of optical fiber is excluded from the simulations. The other two physical processes are simulated based on the Monte Carlo Geant4 toolkits [20,21,22,23,24,25,26,27,28] to provide a reference for developing streak cameras.

## 2. Modeling of Imaging and Analyzing

The streak tube is the core component for achieving ultrafast diagnostic function, and its performance determines the spatiotemporal resolutions, dynamic ranges, spectral response ranges, and detection sensitivity. For a typical streak tube, the laser or X-ray irradiates the photocathode to generate the secondary electrons. And these electrons are accelerated and deflected in an electronic optical system to hit the screen of the streak tube and form a streak image. Typically, images are recorded by a CCD situated behind the screen. The phosphor screen is hit to generate photons, usually in visible light, which then transmitted through the optical fiber panel and illuminate the CCD camera, where the photoelectric effect occurs to produce photoelectrons.

In this work, the imaging process of a streak camera is modeled and simulated using Geant4-based Monte Carlo simulations. Therefore, we made the following flow chart and explained the Monte Carlo simulation process.

(1)Run and Event: These classes relate to the generation of the event, the interface of the event generator, and any secondary particles produced. Their main purpose is to provide the trace manager with particles to track. Each simulation is called a Run, which contains a number of emission particles, and the whole process of each particle and its secondary particles from generation to annihilation or ejection of the reaction area is called an Event.(2)Tracking and Track: Particle simulation in Geant4 is realized through step size and corresponding reaction and energy. Track class is used to analyze the factors limiting steps and corresponding physical processes. In Geant4, each Event contains a primary particle and several secondary particles. The movement trajectory of a single particle and a series of reactions occurring are called a Track, and each Track is divided into several steps according to the Step size. In Geant4, physical reactions occur only in each Step of the particle.(3)Geometry and Magnetic Field: These classes govern the geometry definition of the probe (solid modeling) and the distance between the geometry and the magnetic field in the space where all the geometry is located. Geometry in Geant4 is divided into solid, logical volume and physical volume. Solid contains the three-dimensional information of geometry. Logical volume adds material information on the basis of the solid. Physical volume adds position information on the basis of the logical volume, that is, the position and direction of geometry in the coordinate system. Only physical volumes are modeled by Geant4.(4)Particle Definition and Material: These two classes are used to build particle and material information. Geant4 has a variety of particle, element and material library files that can be directly invoked; the kinetic energy and position of primary particles can be set by the user; and the material of the entity can refer to the material library being defined or the material library not being defined, which can be invoked according to the composition of the material element by setting the material density to generate the corresponding material data.(5)Physics: This class manages all the reaction physical processes in Geant4. Each physical process corresponds to the reaction of certain particles and materials under certain conditions. Users can call the corresponding physical processes according to the needs of simulation, or directly reference the created classes in Geant4 (electromagnetic, hadronic, electromagnetic, etc.), transportation, decay, optical, photolepton_hadron and parameterization).(6)Hit and Digitization: These two classes are used for generating and digitizing particle hit detector examples. In Geant4, a particle colliding with a detector produces a reaction that is recorded as a hit, which is created and managed in different ways, and is collected and stored in a data system for use by the user.(7)Visualization: This class is used to manage the visualization of the physical volume entity, the visualization of the primary particle generation, motion trajectories, and hits, and the generation, motion trajectories of the secondary particles. Visualization engines such as OpenGL, Qt and Open Inventor (for X11 and Windows), DAWN, Postscript (via DAWN) and VRML can be developed in Geant4 (See Figure 1).

### 2.1. Modeling of Imaging in Phosphor Screen

With the increasing demand for high-definition imaging, the research on P43 phosphors is widely conducted in X-ray imaging and cathode ray luminescence fields due to its higher resolution and superior comprehensive performance. Hence, P43 phosphor is selected in this study for the high performance of imaging requirements. The fluorescent powder’s performance directly affects the phosphor screen’s performance and the entire image tube. To better simulate the luminescence of the phosphor screen, it is necessary to first determine the luminescence characteristics of the fluorescent powder used. Some characteristics of P43 phosphors after being made into a phosphor screen are shown in Table 1 [29], such as the luminous efficiency, resolution, and main peak wavelength of the spectrum when the phosphor screen is made of fluorescent powders and excited by 5 keV electrons.

In this work, the response of the phosphor screen is simulated with the Geant4 Monte Carlo toolkit in the case of electron beams orthogonally hitting the phosphor screen. An ideal detector behind the screen records all the impacts of the optical photons coming out of the phosphor. The simulated processes considered in our model with Geant4 version 10.4 are multiple scattering, ionization, atomic de-excitation by fluorescence and optical absorption and scattering for the visible photons.

The optical processes (reflection, refraction) at the interface between two media are not considered. The phosphor is seen as a homogeneous medium with macroscopic optical absorption and scattering coefficients to simplify the simulation. In this work, simulations are based on P43 phosphor, with a few adjustments (related to the screen structure and material) to any kind of phosphor screen.

The influence of the following setup parameters was studied using simulation to optimize the model, such as the monochromatic energy of the electron source and the position distribution of the electron source with a user-defined shape.

Beams of 1 keV, 5 keV, 10 keV, and 15 keV are used here. It is worth noting that the user is free to set the position distribution of the beam to a uniform or Gaussian distribution. The standard deviation of the beam distribution with Gaussian distribution can be refined with respect to the width of each strip beam. In this study, each beam is set to a fringe shape and Gaussian distribution.

In Geant4 toolkits, materials are user-defined based on the types and proportions of elements in the material. At the same time, the program calculates the probability of collisions and reactions between particles and materials based on the material density. The chemical formula of P43 fluorescent powder is Gd_2_O_2_S: Tb, with a density of 4.5 g/cm^3^. The photon energy emitted from the phosphor screen is relatively low, which belongs to the ultra-low energy range in the Geant4 code. On the other hand, to achieve the luminescence of the phosphors screen, corresponding optical properties need to be added to the material, such as fluorescence wavelength and yield.

When establishing the simulation model, it is necessary to create a simulation area, which is usually called “word” in Geant4. All entities and reactions will occur in the scope of word space, and particles beyond the scope of word space will be considered as the end of the simulation. Thus, a simulated background environment in word named “Envelope” should be created: the simulated background environment should be set as a vacuum, and the materials in the background environment can directly call the “vacuum materials” in the material library. In the modeling process, it is necessary to determine the physical size and material of the physical entity modeling. The size of the fluorescent screen is set to a square of 5 cm × 5 cm, and the geometric center of the fluorescent screen is placed at the origin position of the word. According to the electronics industry standard of the People’s Republic of China—phosphor for semiconductor light-emitting diodes, the chemical formula of P43 phosphor is Gd2O2S: Tb, and the density is 4.5 g/cm^3^. The fluorescent screen emits photons with low energy, which belongs to the category of ultra-low energy in common physical reactions, and the reaction model of the program cannot be used. In order to achieve screen luminescence, it is necessary to add corresponding optical properties to the material. After adding the material definition, you can continue to add optical properties to the material.

According to Table 1, the main peak wavelength of P43 fluorescent powder is 556.6 nm, and its corresponding photon energy E = 2.228 eV can be calculated from E = hc/λ. The fluorescence yield in Geant4 code can be calculated based on the conversion relationship between lumens and power of light, as follows:(1)$${\;d\Phi }_{v}=K_{m}V\left(\lambda \right){d\Phi }_{\lambda }\left(lm\right)$$
where ${d\Phi }_{v}$ is the luminous flux in watts, $\;d\Phi_{\lambda }\left(lm\right)$ is the radiant flux in lumens, $V\left(\lambda \right)\;$ is the visual function representing the sensitivity of the human eye to different wavelengths of light, and ${\;K}_{m}$ represents the proportionality constant between the luminous flux and power of different wavelengths of light. For the light with a wavelength of 556.6 nm, 1 lm = 0.00146 W. According to the luminescence efficiency of P43 phosphor at 13.2 lm/W in Table 1, the light luminescence is set to 8.64/keV.

The class G4EmStandardPhysics in Geant4 toolkit is available for universal electromagnetic physical process reference. Due to the low energy of photons generated in the processes of high-energy electron beam exciting fluorescence in phosphor P43, a Geant4 class G4EmStandardPhysicsSS is suitable for this low-energy physical model. Meanwhile, the Geant4 class G4OpticalPhysics and G4PhotoElectricEffect are adopted for optical physical processes and photoelectric effects, respectively.

Output data in Geant4 code can be saved using Ntuple, and the output file is in CSV format. Ntuple combines each data class into a column and then merges each into a table format. During the simulation, two Ntuples are created to store the information for electrons and photons, respectively, whilst the storing methods are different for the two particle types. Information for the electrons needs to be counted till the phosphor screen is reached, when the information for the photons can be saved as soon as being generated. As photons are generated by electrons, recording the electron number by which the photon is generated is necessary. When the simulation of all electrons is completed, the information for all electrons and photons is recorded separately in the corresponding Ntuple.

To verify the luminescence efficiency of P43 fluorescent powder, the total energy of electrons and photons during the simulation process is statistically analyzed, and the luminescence efficiency can be calculated according to Equation (1). The incident electron source is set as a point emission source, with four groups of electron energies of 1 keV, 5 keV, 10 keV, and 15 keV, each emitting 1000 electrons at a time. The energy statistics of electrons and photons at the completion of the simulation are shown in Table 2.

### 2.2. Spatial Resolution of Phosphor Screen

In this model, the phosphor layer thickness is 1 mm, and the chemical composition is gadolinium (Gd): 82.27%, oxygen (O): 8.45%, sulfur (S): 8.45%, terbium (Tb): 0.83%. The fluorescent screen and CCD are coupled through the optical fiber panel, the optical image of the fluorescent screen is transmitted through the optical fiber to the light sensor of the CCD to be recognized, and finally, the optical image of the measured object is output in the CCD camera. Due to the small size and large number of optical fibers, modeling and simulating optical fiber panels require a lot of computer resources in Geant4, and the process of optical fiber panels transmitting optical images is relatively stable. Therefore, physical modeling and simulation of optical fiber panels are not carried out in the simulation, and only functional modeling of optical fiber panel coupling function is carried out in MATLAB9.5. The method of mathematical calculation simulates the function of optical fiber transmission optical image.

With efforts to test the resolution of the phosphor screen, stripe-shaped electron beams with different spatial distributions are used to hit the screen, and the spatial distribution of fluorescence on the screen is analyzed. The photon distribution on the phosphor screen is shown in Figure 2, using striped electron beams with resolutions of 60 lp/mm, 95 lp/mm and 120 lp/mm. The simulation results show that the incident electron beam of 60 lp/mm can be clearly distinguished by the phosphor screen. Although the photon distribution is closer for the electron beam of 95 lp/mm, its boundary can still be detected, regardless of being closer to the limit. However, for the electrons of 120 lp/mm, three sets of photon streak boundary overlap and cannot be distinguished, indicating the phosphor screen cannot meet the resolution of 120 lp/mm.

### 2.3. Modeling of Imaging in CCD Sensor

CCD cameras comprise many components, including a CCD sensor, optical components, an analog-to-digital converter, signal processor, storage medium, interface and controller. The main component related to imaging results is the CCD sensor, which is simultaneously the core component of a CCD camera, with parameters directly affecting the performance and imaging results of the CCD camera.

In this work, the CCD sensor’s response is simulated with the Geant4 Monte Carlo toolkit for fluorescent photons emitting from the phosphor screen orthogonally impinging onto the CCD sensor. The main simulated processes taken into account in our model with Geant4 version 10.4 are the photoelectric effect for the fluorescent photons and the optical absorption and scattering for the photons. Thus, a simulation model is established to analyze the photoelectric effect occurring in CCD sensors. When photons generated in the phosphor screen illuminate on the CCD sensor, some photoelectrons with different energies will be generated. The CCD sensor converts the photoelectrons in each pixel into voltage or current signals, and the signals’ magnitude in each pixel represents the pixel’s brightness. Note that only the photosensitive material of the CCD sensor is considered for generating photoelectric effect and establishing the simulation model in Geant4, while others have none. The thickness of the photosensitive material in CCD sensor is usually between a few micrometers and a dozen micrometers; thus, its thickness is set to 10um in Geant4.

CCD sensors use silicon as the substrate and are doped with a specific concentration of element B to form P-type silicon. In Geant4, the photosensitive material is defined by the mass ratio of silicon and boron elements. Also, the photoelectric effect is used to model and analyze the physical process of photoelectrons generated by fluorescence irradiation on CCD-photosensitive materials. Due to the millions of pixels in the CCD sensor, the program will occupy a large amount of memory during simulation if each pixel is individually constructed as a solid in Geant4, leading to extremely high requirements for computer performance. The CCD photosensitive material is established as a whole solid in the modeling process to be more effective regarding resources and simulation time. Meanwhile, the pixel’s position, where the photoelectrons are generated in each reaction, is determined by the position where the photoelectric effect occurs in the photosensitive material irradiated by photons. Then, the charge amount of photoelectrons in each pixel is summarized.

CCD sensors use silicon as a base, with a certain concentration of boron-doped P-type silicon: the boron-doping concentration used is 10^19^/cm^3^, so the mass of boron accounts for about 0.07%, and can be defined by the mass ratio of silicon and boron to define the photosensitive material.

In Geant4 PhysicsList, the class G4EmStandardPhysics_option3 is adopted to analyze low-energy electromagnetic reactions, and the class G4OptialPhysics is for analyzing optical processes. Due to the low energy of photons in the simulation, it is necessary to adjust the minimum reaction energy value to the minimum other than 0.

Referring to the parameters of the state-of-the-art CCD sensor KAF-50100, with a single pixel of 6 μm × 6 μm and 500,000 pixels, the highest resolution that can be reached is 83 lp/mm. During the simulation, the energy and position information of the photoelectric effect are recorded. Considering the presence of shot noise and dark current noise during the imaging process of a CCD camera, a noise reduction measurement is adopted in the data processing after simulation to more accurately display the final output image of the streak camera.

### 2.4. Spatial Resolution of CCD Camera

To verify the resolution of the CCD camera, the shape of the photon emission source is set to three sets of rectangular streaks (note: the photons here are not generated on the fluorescent screen, but manually set more regular photon streaks, mainly for testing the resolution of the CCD sensor), and simulation measurement is conducted with streaks with resolutions of 30 lp/mm, 60 lp/mm, and 90 lp/mm, respectively. The simulation results are shown in Figure 3. For an incident photon streak of 30 lp/mm shown in Figure 3a, the boundaries of the three sets of streak images displayed on the CCD camera displayed in Figure 3b can be clearly distinguished, and the overall brightness of the three sets of streaks is consistent. At the same time, noise can be observed on the pixels without illumination in the image. For the 60 lp/mm morning photon streaks, the three sets of streaks in the imaging can still be clearly distinguished as illustrated in Figure 3c. For 90 lp/mm photon streaks, the three sets of streak boundaries in imaging overlap and cannot be distinguished as shown in Figure 3d, indicating that the CCD camera cannot resolve 90 lp/mm.

### 2.5. Co-Simulation and Modeling of Imaging of Streak Camera

To verify the process from the output electrons of the streak image tube to CCD imaging, it is necessary to simulate the entire physical process, including how electrons generate photons on the phosphor screen and how photons illuminate the CCD sensor. Additionally, to test the imaging effect of electron beams with different spatial distributions as shown in Figure 4a, multiple sets of imaging characteristics of electron beams with different distributions were simulated.

The co-simulation results are shown in Figure 4. The position distribution of the electron beam is seen in Figure 4a, with a streak spacing of about 16 μ m, corresponding to a resolution of 60 lp/mm. The electron density of each streak varies with the Y-axis, indicating that the electron density evolves over time. Figure 4b shows the fluorescence distribution generated on the P43 phosphor screen, similar to the electron distribution. When the photons with this distribution are irradiated onto the CCD sensor, the imaging results on the CCD camera are finally expressed in Figure 4c after simulation by the photoelectric effect and considering the noise. The contours of the three sets of streaks can be distinguished. It is observable that the brightness change of each set keeps well in agreement with the distribution of electrons from the streak image tube, signaling that the simulation results can correctly reflect the electron distribution by the streak image tube, thus realizing the simulation of the streak camera’s imaging process.

## 3. Conclusions

This work investigates the physical imaging process as electron beams bombardment on a fluorescent screen, and implements the simulation modeling and analysis. The fluorescent powder P43 is selected, and a Monte Carlo Geant4 toolkit is adopted for modeling the physical process of fluorescence generated by electron bombardment on the fluorescent screen. The fluorescence luminescence efficiency and spatial resolution of the P43 fluorescent screen are simulated using relevant classes of the Geant4 toolkit, of which the results show good agreement with the fluorescent material’s parameters, and the resolution ability of the selected fluorescent powder and the reliability of the simulation method can be then verified. Simultaneously, the process of how photoelectrons are generated by fluorescence irradiation on the CCD sensor is modeled and simulated. The selected CCD sensor’s resolution ability is verified considering the influence of noise and fluorescence irradiation with different spatial resolutions. Finally, a co-simulation is carried out on the entire imaging process of the fluorescent screen and CCD sensor. Several sets of electron beams with different spatial distributions are used to bombard the fluorescent screen to generate photons and irradiate the CCD sensor to produce photoelectrons and images. These analysis methods and simulation results provide valuable references for streak camera imaging, weak signal intensifier imaging, X-ray tube imaging, etc., promoting the development of streak cameras and other imaging instruments, as well as in fields such as ultrafast diagnosis and having significant application values in related science fields.

## Figures and Tables

**Figure 1 sensors-25-00881-f001:**
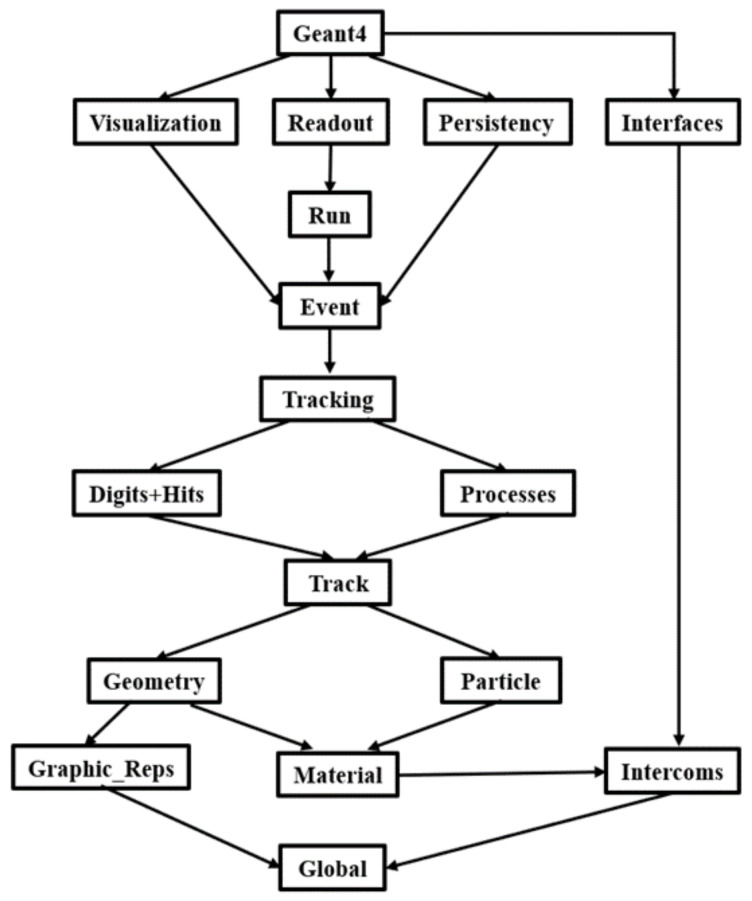
Geant4 architecture block diagram.

**Figure 2 sensors-25-00881-f002:**
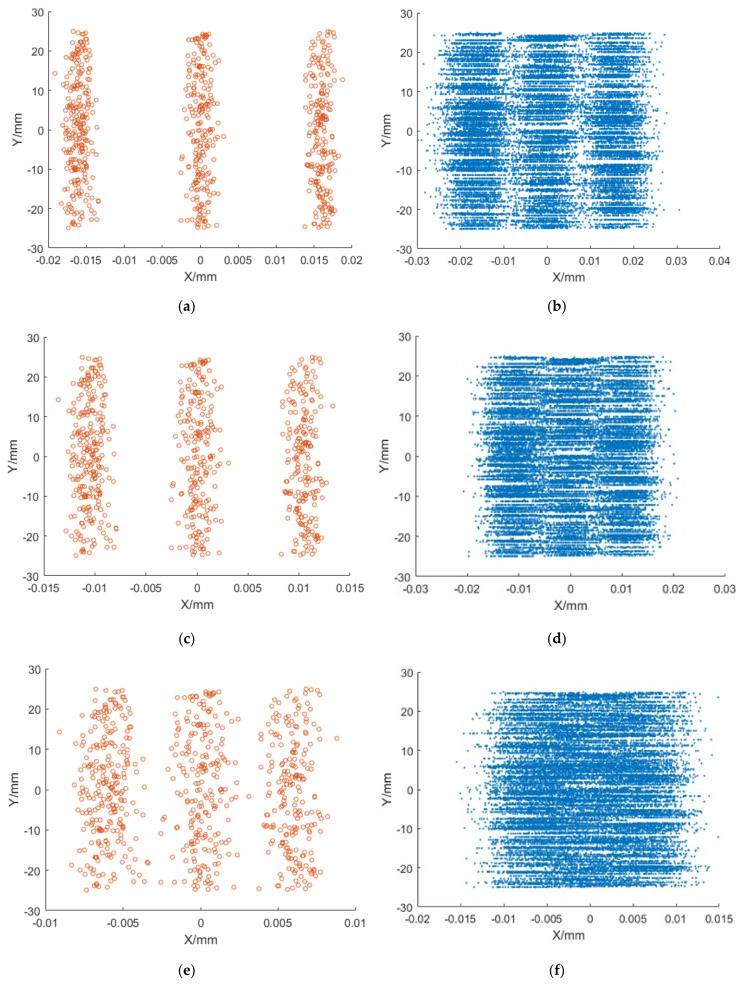
Simulation results of phosphor screen resolution. (**a**) 60 lp/mm electron beam; (**b**) photon distribution corresponding to 60 lp/mm electron beam; (**c**) 95 lp/mm electron beam; (**d**) photon distribution corresponding to 95 lp/mm electron beam; (**e**) 120 lp/mm electron beam; (**f**) photon distribution corresponding to 120 lp/mm electron beam.

**Figure 3 sensors-25-00881-f003:**
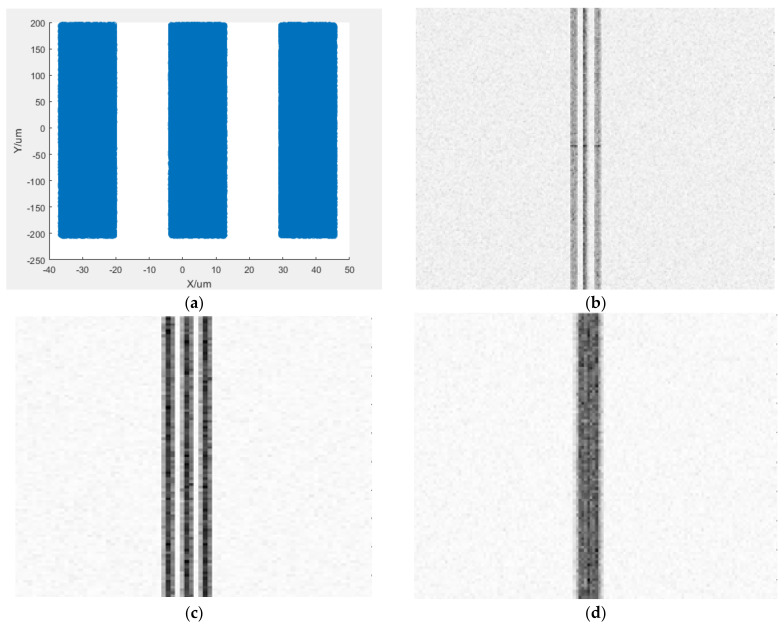
Simulation results of CCD imaging. (**a**) photon distribution of 30 lp/mm; (**b**) CCD imaging for photon of 30 lp/mm; (**c**) CCD imaging for photon of 60 lp/mm; (**d**) CCD imaging for photon of 90 lp/mm.

**Figure 4 sensors-25-00881-f004:**
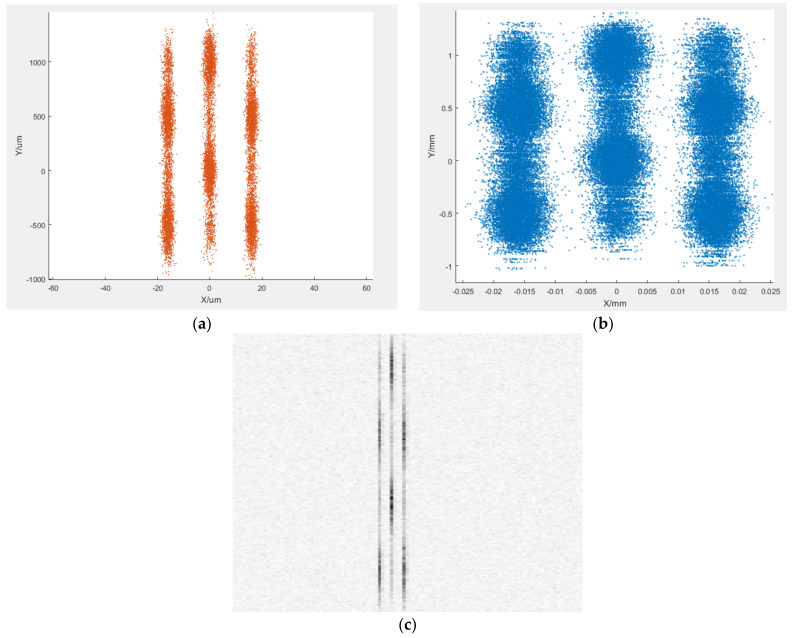
Simulation results of streak camera imaging. (**a**) distribution of output electrons from streak image tube; (**b**) distribution of incident photons on CCD; (**c**) imaging results on CCD camera.

**Table 1 sensors-25-00881-t001:** Performance of P43 phosphor at 5 keV.

Parameter	P43
luminous efficiency/(lm/W)	13.2
resolution/(lp/mm)	95
Main wavelength/nm	556.6

**Table 2 sensors-25-00881-t002:** Luminous efficiency vs. electron with different energy.

Energy of Electron Beam	Total Energy of Electron Beam	Total Energy of Photons	Luminous Efficiency
1 keV	1 MeV	17.7 keV	12.12 lm/W
5 keV	5 MeV	99 keV	13.56 lm/W
10 keV	10 MeV	179 keV	12.26 lm/W
15 keV	15 MeV	267 keV	12.20 lm/W

## Data Availability

The data that support the findings of this study are available from the corresponding author upon reasonable request.

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
