# Peer review of "Simulation and Analysis of Imaging Process of Phosphor Screens for X-Ray Imaging of Streak Tube Using Geant4-Based Monte Carlo Method"

_sensors, 2025, doi:10.3390/s25030881_

Round 1

Reviewer 1 Report

Comments and Suggestions for Authors

Ultrafast diagnostic technologies, including streak cameras, can help to know more mechanisms of laser interaction with matters. So they are important in the particle accelerator, inertial confinement fusion. In this paper, authors studied the properties of electron beams on the phosphor with parameters of CCD and P43 and Monte Carlo simulation, Geant4. The results might be helpful to estimate (understand) the properties of electron beams from the screen of CCD, but the present form of paper does not reach the standard to publish by Sensors.  

1.      The authors used Monte Carlo Geant4 as a main tool to simulate the process of imaging. This software is important to the work of this paper. I suggest the authors add some description and examples of applications of the Geant4 in the Section 2.

2.      The Sections 2.1 and 2.3 are the models of the imagine in phosphor screen and CCD. But there are not enough physical formulas to describe this process, only with Equation (1). Authors should consider add some formulas to describe the models.

3.      The labels in the Figs. 1-3 are not large enough.

4.      As a simulation paper, authors should be easier to obtain much data with less cost. The results shown in the Figs. 1-3 are not enough to obtain useful conclusions. I suggest authors to carry out more simulations to investigate the detail of the imagine, such as the width of electron beams on the imaging.  

Reviewer 2 Report

Comments and Suggestions for Authors

The reviewed paper "Simulation and Analysis of Imaging Process of Phosphor Screens for X-ray Imaging of Streak Tube Using GEANT4-based Monte Carlo Method" investigates the processes in the final part of streak cameras, specifically focusing on the phosphor+CCD detector module. The authors employ the well-established Geant4-based Monte Carlo simulation method to model the electron-photon processes in solid materials. The paper can be divided into two main sections: (1) the investigation of phosphor fluorescence induced by keV-electron beams and (2) the simulation of photoelectron responses in CCD pixels triggered by photons originating from the phosphor.

This type of research is potentially of interest, but I have several significant remarks and recommendations for the authors that should be addressed before publication:

1. The term “imaging” is used loosely throughout the paper to describe physical processes in phosphor and CCD materials. Imaging typically involves the conjugation of an object and its image, which does not seem to occur in this context. A more appropriate term, such as “imaging processes,” should be used to better reflect the scope of the study.

2.  The approach to fluorescence emission modeling presented in Subsection 2.1 is unclear. The rationale for introducing luminous efficiency and Eq. 1, given the use of Geant4 algorithms for simulating electron interactions with atoms, is not well-explained. Fluorescence emission events should ideally be simulated as part of the electron scattering and dissipation processes.

3. I recommend the authors include a flowchart detailing their calculation and Monte Carlo simulation processes for better clarity.

4. The concept of quantum efficiency (photon yield per electron) is more straightforward and commonly used in Monte Carlo simulations. Eq. 1, which appears to reference an ordinary light source and human-eye spectral sensitivity, is irrelevant in the context of electron-matter interactions. Using watts and lumens as units is also unconventional for this type of study.

5. The parameters presented for P43 phosphor in Table 1 (referred to incorrectly as "Tab 1") require validation, with data sources explicitly cited. Additionally, the table title (“performance of commonly-used phosphors”) is ambiguous and needs revision.

6. The fluorescence efficiency of phosphors depends heavily on electron energy. Values presented for 5-keV electrons cannot be assumed to apply to other energies.

7. The discussion in the fourth paragraph (lines 105–109) on page 3 is unclear and requires better explanation.

8. Table 2 seems inaccurate, as luminous efficiency values typically vary significantly with electron energy. Reference data (e.g., https://www.proxivision.de/datasheets/image-intensifier-general-information.pdf) should be consulted. Additionally, the term “Total energy of photo” should be clarified.

9. Key details necessary for simulating luminescence processes in the phosphor are missing. These include:  phosphor layer thickness,  chemical composition (e.g., the partial gadolinium content in P43),  substrate material, distance between the phosphor layer and the CCD detector.

10. In Subsection 2.3) the authors do not discuss the potential impact of the technological, geometrical, and electrical structures of the CCD camera, which could also be analyzed using Geant4 simulations.

11. Page 5, line 189: What is the mass ratio of silicon and boron in the CCD sensitive layer? This information is crucial for accurate simulations.

In conclusion, while the paper addresses an intriguing topic, the aforementioned issues significantly hinder its clarity and reliability. I recommend that the authors thoroughly revise the manuscript, addressing these remarks to improve its scientific rigor and clarity before publication.

Round 2

Reviewer 1 Report

Comments and Suggestions for Authors

now it can be published by Sensors.

Reviewer 2 Report

Comments and Suggestions for Authors

None